# Radiation Hormesis to Improve the Quality of Adult *Spodoptera litura* (Fabr.)

**DOI:** 10.3390/insects13100933

**Published:** 2022-10-15

**Authors:** Neha Vimal, Nilza Angmo, Madhumita Sengupta, Rakesh Kumar Seth

**Affiliations:** Applied Entomology and Radiation Biology Lab, Department of Zoology, University of Delhi, Delhi 110007, India

**Keywords:** radio-genetic pest control tactics, inherited sterility technique, sterile insect technique, cutworm, low-dose ionizing radiation, antioxidant genes, survivorship curves

## Abstract

**Simple Summary:**

*Spodoptera litura* (Fabr.) is a serious noctuid lepidopteran pest in India that can be managed using the eco-friendly, radio-genetic inherited sterility (IS) technique. Improving the quality of laboratory-reared moths might increase the efficacy of released sterile moths using this technique. In the present study, radiation hormesis using low-dose ionizing radiation administered to different ontogenetic stages was studied. The growth potential and survival of the developing stages derived from these treated stages was assessed, followed by an assessment of the expression profile of longevity- and viability-related genes. The findings indicated that ionizing radiation doses of 0.75–1.0 Gy might be used as hormetic treatments in eggs, larvae, or the pupal stages of *S. litura*. This would result in more viable and competitive adult moths for use in the IS technique.

**Abstract:**

Mass rearing of insects of high biological quality is a crucial attribute for the successful implementation of sterile insect release programs. Various ontogenetic stages of *Spodoptera litura* (Fabr.) were treated with a range of low doses of ionizing radiation (0.25–1.25 Gy) to assess whether these gamma doses could elicit a stimulating effect on the growth and viability of developing moths. Doses in the range of 0.75 Gy to 1.0 Gy administered to eggs positively influenced pupal weight, adult emergence, and growth index, with a faster developmental period. The enhanced longevity of adults derived from eggs treated with 0.75 Gy and 1.0 Gy, and for larvae and pupae treated with 1.0 Gy, indicated a hormetic effect on these life stages. Furthermore, the use of these hormetic doses upregulated the relative mRNA expression of genes associated with longevity (*foxo, sirtuin 2 like/sirt1, atg8*) and viability/antioxidative function (*cat* and *sod*), suggesting a positive hormetic effect at the transcriptional level. These results indicated the potential use of low dose irradiation (0.75–1 Gy) on preimaginal stages as hormetic doses to improve the quality of the reared moths. This might increase the efficiency of the inherited sterility technique for the management of these lepidopteran pests.

## 1. Introduction

*Spodoptera litura* (Fabr.) is a polyphagous noctuid pest which infests almost 150 different plants belonging to 44 families all over the world, and in India alone it affects almost 60 plant species [1]. In the past, chemical insecticides have been used extensively to manage populations of this species. This, however, has resulted in increased resistance to these chemicals, and their indiscriminative use has become a major problem for human health and the environment [2]. Over the last few decades, ionizing radiation has been used in specific sterile insect release programs to manage populations of several lepidopteran pests [3]. The sterile insect technique (SIT) involves the use of 100% radio-sterilized males to mate with wild females, which results in no offspring. In the case of the Lepidopteran group, the insects are intrinsically highly radio resistant, and a dose that would obtain sterility entails serious somatic damage, leading to reduced longevity and competitiveness. However, an alternative approach, i.e., the inherited sterility technique (IS), is designed to overcome the drawback of SIT by releasing sub-sterile male moths exposed to lower gamma irradiation doses, with inherent capacity to induce enhanced sterility in their F1 generation. A mating of a sub-sterile male with a wild female results in some offspring that are sterile [4]. 

The quality of the mass-reared insects that are destined to be released in sterile insect release programs is very important and crucial for the success of these pest control programs. Sterilizing or sub-sterilizing doses of gamma radiation have been the subject of extensive research, but little attention has been given to the effects of very low radiation doses that induce hormesis in lepidopteran insects. However, the concept of mild stress-induced physiological hormesis has been explored using a low dose of oxygen within the context of the management of some lepidopteran insects [5,6].

Exposing an organism or cell to low doses of physical, chemical, or biological agents can produce positive biological responses, whereas a higher dose of the same agent can decrease the beneficial effects [7]. This nonlinear dose response towards various stressors can be seen across a wide range of organisms, i.e., from bacteria to vertebrates [8]. Hormesis can be defined as a dose-response relationship characterized by a reversal in response between low and high doses of a stressor, thus characterizing a biphasic relationship [9,10,11]. The doses that induce hormetic responses are limited in range and typically below the threshold where no effect is generally noticed [12]. In a wide range of organisms, a low dose of a stressor (e.g., alkylating agents, thermal and oxidative stress, ionizing radiation, chemical agents, heavy metals) induces a wide range of biological effects which may modulate the physiological responses [13].

The concept of radiation hormesis is now widely recognized and accepted as a general stress response phenomenon [9], and in many cases, it has influenced the longevity of an organism. However, the molecular mechanism underlying this change in lifespan due to radiation hormesis is poorly understood [14]. In the round worm *Caenorhabditis elegans* (Nematoda: Rhabditidae), exposure to several low irradiation doses triggered an adaptive biological response which in turn increased the lifespan; this might be related to *sir2, p53* and *foxo* genes [15]. *Sirt1*, a mammalian homolog of yeast *sir2*, has been found to regulate cell survival, replicative senescence, and metabolism by deacetylating histones. It also modulates transcription factors like *p53, foxo, NF-kB* and pgc-1α [16,17,18]. Sirt1 has dual role in response to oxidative stress on the function of *foxo* gene; it induces cell cycle arrest and resistance to oxidative stress but inhibits the *foxo*’s capacity to cause cell death [19]. Additionally, radiation-induced hormesis and an adaptive response have been reported to be induced by the gene *p53* in *Drosophila melanogaster* (Diptera: Drosophilidae) [20].

It has also been shown that DNA damage sensors, such as *atm* and antioxidant genes such as superoxide dismutase (*sod)* and catalase (*cat*), also play a key role in detoxification after exposure to different stressors like oxidative stress, ionizing radiation and UV radiation. Additionally, autophagy related genes (*atg*) are known to affect the longevity of an organism [21,22,23,24,25,26,27]. Lifespan prolongation via hormesis can also be induced by targeting antioxidant genes like *cat* and *sod*, which may contribute to the detoxification of reactive oxygen species (ROS), repair of damaged DNA, and cell survival [28].

The IS technique has been proposed as a potential tactic for the management of *S. litura* [4,29,30]. In the present study, an attempt was made to assess the effect of a range of very low doses of gamma radiation on the development and longevity of *S. litura*. The potential hormetic doses of radiation were administered to different ontogenetic stages, and their effect on longevity- and viability-associated factors/genes like *foxo, sirtuin 2 like/sirt1, atm, atg8, p53, cat* and *sod* was assessed. 

## 2. Materials and Methods

### 2.1. Culture and Maintenance of Spodoptera litura

Adult moths of both sexes were obtained from agricultural fields around Delhi, India, and the colony was maintained in an experimental facility at a temperature of 27 ± 2 °C with 75 ± 5% relative humidity (RH) and an equinox photoperiod of 12 h light: 12 h darkness provided by compact fluorescent lamps (480–570 nm in wavelength). Care was taken to avoid microbial infection in the culture [31]. Adult moths were kept in 20 cm × 20 cm × 20 cm Perspex–nylon net cages for mating and were provided ad libitum with 10% honey solution in cotton balls. Four pairs of adult moths were paired in one cage. After mating and oviposition, the eggs were collected and placed in a plastic container (Tarson, Kolkata, West Bengal, India, 10 cm diameter × 12 cm height). The freshly hatched larvae were reared on a semi-synthetic diet until pupation [4], and the rearing cycle continued. 

### 2.2. Irradiation Treatments

For the experiment, 0–1 day old eggs, 0–1 day old early third instar larvae (L3), and 2–3 day old pupae were exposed to a range of low irradiation doses (0.25–1.25 Gy) in a Co^60^ Tele therapy unit (Bhabhatron-II, Panacea Medical Technologies Pvt Ltd., Bengaluru, India) at the Institute of Nuclear Medicine and Allied Sciences (INMAS) of the Defense Research and Development Organization (DRDO), Delhi, India. Non-irradiated insects in the corresponding ontogenetic stages of the respective regimen were used as control groups. All the stages were irradiated in Petri dishes (90 mm) at 80 cm source to surface distance (SSD) and with a field size of 35 × 35 cm. The radiation doses were administered at the inner surface of the Petri dish.

The irradiator beam was calibrated following IAEA’s TRS-398 protocol, using a calibrated Farmer type, 0.6 cc-volume ionization chamber in water phantom. During the experimental phase, the measured dose rate ranged from 1.09 Gy/min to 0.74 Gy/min. The dose uniformity was better than ±3% within the central 80% of the field and the overall output calibration uncertainty was within ±2%.

### 2.3. Development

The development of irradiated and control eggs and larvae were assessed by the following parameters: percent pupation, weight of the pupae, and percentage of adult emergence. In the case of the irradiated pupae group, adult emergence and pupal period were the parameters assessed. The growth index [32] was calculated as the ratio of percentage adult formation and development period (days). The development period of the different experimental groups was defined as the time elapsed between the irradiation event and adult emergence. 

In the experimental group of irradiated eggs, each replicate consisted of 10 cohorts, with each cohort consisting of 60–70 eggs. The average pupal weight and development period of 10 insects from each cohort was considered as one replicate. The irradiated larvae (L3) experimental group was replicated 10 times, and each replicate consisted of a cohort of 50–60 third instar larvae. The average pupal weight and development period of 10 insects from each cohort was considered as one replicate. The irradiated pupae experimental group was replicated 10 times, and a cohort of 25–30 pupae (2–3 days old) was considered as one replicate. The average pupal weight and development period of 10 insects from each cohort was considered as one replicate.

### 2.4. Longevity

After emergence, the virgin male and female moths were separated and placed in different cages (20 cm × 20 cm × 20 cm) under ambient environmental conditions (27 ± 2 °C, 75 ± 5% RH, and 12 h light: 12 h darkness), with a cotton wool ball soaked in 10% honey solution. The survival of the moths was recorded daily, and the longevity of virgin male and female moths kept separately was also assessed. The average longevity from each cohort of 10 virgin moths (male and female in each experimental group) with respect to dose and treated ontogenetic stage constituted one replicate. The experiment was replicated 10 times.

For assessing the longevity of mated moths, 10 pairs of moths were placed in one cage (60 cm × 60 cm × 60 cm) for one replicate and experiment was repeated 10 times.

### 2.5. Tissue Collection for qPCR

Samples of brain and fat tissues were collected after dissection in Belar’s saline from 0–1-day-old moths of both sexes derived from the different irradiated development stages (eggs, L3 larvae, or pupae) one hour after the lights went off in the scotophase. The control non-irradiated moths were maintained under the same conditions. For each variant, six biological replicates were sampled (tissue from one moth constituting one replicate). The tissues were stored in Trizol reagent at −80 °C in an eppendorf.

### 2.6. RNA Extraction and cDNA Synthesis

Total RNA was isolated from the samples using the phenol-chloroform extraction method [33]. The RNA concentration was determined using a NanoDrop ^TM^ 2000C (Thermo Fischer Scientific, Wilmington, DE, USA). An A260/280 ratio of the RNA samples of 1.8–2.0 was considered pure RNA. Single-stranded cDNA was synthesized using 1µg of total RNA pretreated with DNase1 (Promega, Madison, WI, USA), hexanucleotide primers, and reverse transcriptase of cDNA kit (Thermoscientific, K1622, Waltham, MA, USA). Afterwards, the amplification of cDNA was conducted with a reference gene, i.e., β actin, and the quality of cDNA was checked on 1% agarose gel.

### 2.7. Measurement of Gene Expression of Candidate Genes

For the measurement of mRNA expression of genes, gene-specific primers were developed from the cDNA sequences using the Eurofins MWG Operon primer design online program (http://www.operon.com/tools/oligo-analysis-tool.aspx) (accessed on 10 December 2020) and are included in Table 1. The mRNA expressions were conducted using a qualitative real-time PCR (qPCR) technique in thermocycler (ViiA7, Applied Biosystems, Foster City, CA, USA) by using primer sequence (Table 1) and SYBR Green PCR Master Mix. The relative mRNA expression was calculated as fold change: the 2^−(ΔΔCt)^ value [34]. Both sample and reference (EF1α) gene were run in duplicates in a reaction volume of 6 µL (1 µL of cDNA + 1 µL of forward primer + 1 µL reverse primer + 3 µL of Power SYBR Green Master Mix). The fluorescence exceeding the background level provided the cycle threshold (C_t_), which was used to calculate ΔC_t_ (C_t target gene_ − C_t reference gene_) values. Then, each ΔC_t_ value was normalized using the ΔC_t_ value of the pool sample, which consisted of a mix of cDNA of all samples and determined ΔΔC_t_ values, and its negative value powered to 2 (2^−ΔΔCt^) was plotted.

### 2.8. Statistical Analyses

The mean longevity of the different experimental groups that were irradiated with different low doses of gamma rays was statistically analyzed by one-way ANOVA followed by Tukey’s post-hoc test. To estimate the survival functions, the Kaplan–Meier method was used, and survivorship curves were plotted which showed the probability of survival changes over time [35]. Statistical significance was determined using the log-rank (Mantel-Cox) test for each condition. Statistical significance was considered to be *p* < 0.05. 

One-way ANOVA was also used to test the effects of possible hormetic gamma doses on various growth parameters and gene expression, followed by Tukey’s test to detect differences between the groups at *p* < 0.05. All statistical analysis was carried out using the GraphPad prism software program, version 9.3.1 (San Diego, CA, USA). 

## 3. Results

### 3.1. Low-Dose Ionizing Radiation and Development Parameters

The effect of low doses of irradiation on eggs was assessed based on pupal weight, percentage pupation, percentage adult emergence, development period (days), and growth index. A dose of 0.75 and 1.0 Gy resulted in a significant increase in pupal weight (*F*_(5,54)_ = 16.99, *p* < 0.001) and percentage adult emergence (*F*_(5,54)_ = 9.17, *p* < 0.001), whereas percentage pupation only increased significantly with a dose of 1.0 Gy (*F*_(5,54)_ = 10.08, *p* < 0.001) (Table 2a). Irradiation of L3 larvae with 1.0 Gy resulted in a significant increase in percentage pupation (*F*_(5,54)_ = 6.33, *p* < 0.001) and percentage adult emergence (*F*_(5,54)_ = 3.82, *p* < 0.001) (Table 2b). No statistically significant difference was observed in development of pupal stage due to LDIR (adult emergence: *F*_(5,54)_ = 0.76, *p* = 0.58; developmental period: *F*_(5,54)_ = 0.62, *p* = 0.68; and growth index: *F*_(5,54)_ = 0.90, *p* = 0.49 ) (Table 2c). The growth index improved significantly when eggs were irradiated with 0.75 and 1.0 Gy (*F*_(5,54)_ = 7.76, *p* < 0.001), and larvae with 1.0 Gy (*F*_(5,54)_ = 4.29, *p* = 0.002). The irradiation treatment of pupae did not improve any development parameters. 

### 3.2. Low-Dose Irradiation and Adult Moth Lifespan

Exposing eggs to 0.75 Gy and 1.0 Gy of gamma radiation increased the mean adult virgin male lifespan by 16.4% and 22.8%, respectively (*F*_(5,54)_ = 20.45, *p* < 0.001), and the lifespan of virgin female moths by 12.3% and 19.7%, respectively, compared with the controls (*F*_(5,54)_ = 16.89, *p* < 0.001) (Table 3). There was a shift observed in the Kaplan-Meier survivorship curves indicating a significant difference in survivorship percentage due to possible hormetic doses (0.75 Gy and 1.0 Gy) in virgin male moths (log-rank *X*^2^ = 87.03, df = 5, *p* < 0.001) as well as in virgin female moths (log-rank *X*^2^ = 67.94, df = 5, *p* < 0.001) compared to the control (0 Gy), which originated from eggs that had received a low dose irradiation (Figure 1). 

Exposing larvae to a radiation dose of 1.0 Gy increased the mean longevity of virgin males by 24.7% (*F*_(5,54)_ = 17.24, *p* < 0.001) and of virgin females by 22.9% (*F*_(5,54)_ = 18.31, *p* < 0.001), compared with the controls. With regard to survivorship percentage, a significant difference was observed at 1.0 Gy in both virgin males (log-rank *X*^2^ = 69.30, df = 5, *p* < 0.001) and virgin females (log-rank *X*^2^ = 77.70, df = 5, *p* < 0.001).

Similar results were observed with pupae irradiated with 1.0 Gy. Mean longevity was increased in virgin male and female adults by 12.5% (*F*_(5,54)_ = 7.32, *p* < 0.001), and 11.8% (*F*_(5,54)_ = 8.87, *p* < 0.001), respectively, compared with the control. In comparison with the control, there was a significant difference in survivorship percentage observed in the virgin moths derived from 1.0 Gy treated pupae (in virgin males, log-rank *X*^2^ = 31.20, df = 5, *p* < 0.001; in virgin females, log-rank *X*^2^ = 43.37, df = 5, *p* < 0.001)(Figure 1). 

In mated male and female moths irradiated with 0.75 Gy in the egg stage, average longevity increased with 11.7% in males and 11.1% in females, compared with controls (Table 3). Irradiating the eggs with 1.0 Gy resulted in an increase in longevity of 18.2% in mated males and 16.05% in mated females, compared with controls (mated male, *F*_(5,54)_ = 6.84, *p* < 0.001; mated female, *F*_(5,54)_ = 8.07, *p* < 0.001) (Figure 2). Exposure to 0.75 Gy and 1.0 Gy in the egg stage also increased survivorship of mated male moths (log-rank *X*^2^ = 37.50, df = 5, *p* < 0.001), as well as that of mated female moths (log-rank *X*^2^ = 67.94, df = 5, *p* < 0.001), compared to controls (Figure 2).

Exposure of larvae (L3) to low-dose irradiation also increased average longevity by 24.7% in mated male moths and 24.4% in mated female moths (mated males, *F*_(5,54)_ = 9.62, *p* < 0.001; mated females, *F*_(5,54)_ = 16.03, *p* < 0.001), compared with controls. Exposing pupae to low-dose irradiation (1.0 Gy) increased average longevity of mated males by 14.6% and of mated female moths by 12.6% (mated male, *F*_(5,54)_ = 8.78, *p* < 0.001; mated female, *F*_(5,54)_= 6.02, *p* < 0.001), compared with controls. The survivorship curves of the mated adults derived from low-dose irradiation treatment of moths in the larval and pupal stage also indicate better survival of the 1.0 Gy treatment group (in L3 stage: mated males, log-rank *X*^2^ = 39.12, df = 5, *p* < 0.001; mated females, log-rank *X*^2^ = 54.99, df = 5, *p* < 0.001) (in pupa stage: mated males, log-rank *X*^2^ = 34.93, df = 5, *p* < 0.001; mated females, log-rank *X*^2^ = 24.73, df = 5, *p* = 0.002), compared with controls (Figure 2).

### 3.3. Hormetic Gamma Doses and Gene Expression

Male moths responded to potential hormetic gamma doses administered to the various ontogenetic stages with a significant increase in the expression of the *sirtuin 2 like/sirt1* and *foxo* genes in brain (*sirtuin 2 like/sirt1 p* = 0.0003; *foxo*: *p* = 0.0041) and fat tissue (*sirtuin 2 like/sirt1*: *p* = 0.001; *foxo*: *p* = 0.0005). The autophagy gene *atg8* showed a significant increase in the mRNA level in adult male brain (*atg8*: *p* = 0.0174) and fat tissue (*atg8*: *p* = 0.0094). In adult male moths, the expression of the *atm* gene in the brain was significantly reduced (*atm*: *p* = 0.017), but this was not the case in fat tissue. No significant difference was observed in the expression of the antioxidant gene cat in brain tissue, but increased mRNA levels were observed in the fat tissue of males. Expression of the *sod* gene significantly increased in both adult brain (*sod*: *p* = 0.014) and fat tissue (*sod*: *p* = 0.003) of male moths. However, the expression level of the *p53* gene was not significantly different in either brain (*p53*: *p* = 0.081) or fat tissue (*p53*: *p* = 0.077) of the males (Figure 3).

The expression of the *foxo* and *sirtuin 2 like/sirt1* genes in females that were exposed to hormetic gamma irradiation in the different ontogenetic stages was significantly upregulated in brain (*sirtuin2 like/sirt1*: *p* = 0.002; *foxo*: *p* = 0.004). and fat tissues (*sirtuin2 like/sirt1*: *p* < 0.001; *foxo*: *p* < 0.001). However, the expression of the *atm* and *p53* genes did not show any significant difference in either brain (*atm*: *p* = 0.93; *p53*: *p* = 0.67) or fat tissue (*atm*: *p* = 0.3677; *p53*: *p* = 0.8668). The expression of the antioxidant gene *sod* was significantly increased in both brain (*sod*: *p* = 0.0083) and fat tissue (*sod*: *p* = 0.0079) of females, while the upregulation of the *cat* gene was only observed in fat tissue (*cat*: *p* = 0.0057) of females. The autophagy gene *atg8* showed no increased expression in brain tissue (*atg8*: *p* = 0.536), but increased expression was apparent in the fat tissue (*atg8*: *p* = 0.0013) of female moths (Figure 3).

## 4. Discussion

A plethora of research has been conducted in the past on the effect of high-dose irradiation on insects in relation to radiation-based insect pest control strategies [36]. However, our knowledge on the effect of low doses of ionizing radiation on insects is very limited. In the present study, the effect of low doses of ionizing radiation (0.25–1.25 Gy) administered to various ontogenetic (preimaginal) stages (egg, L3 and pupa) of *S. litura* was assessed on growth, survival, and viability.

Irradiating eggs exposed to low doses of radiation (0.75 and 1.0 Gy) resulted in an enhancement of pupal weight, percentage of adult emergence, and growth index. After irradiating larvae with 1.0 Gy, the percentage pupation and adult emergence were positively influenced. However, these parameters were not influenced by low-dose irradiation of the pupae (0.25–1.25 Gy), which might be attributed to the limited period of time after LDIR to pupal stage. Similar results were found by different hormetins: for example, chemical-induced hormesis was observed after exposing larvae of the fungus gnat, *Bradysia odoriphaga* (Diptera: Sciaridae), to Chlorfenapyr, with increased pupal weight as a result [37]; increased percentage pupation was observed in the queen blowfly *Phormia regina* (Diptera: Calliphoridae) when exposed to low levels of heavy metal [38]; and exposing *Trichogramma atopovirilia* (Hymenoptera: Tricogrammatidae)and *T. pretiosum* (Hymenoptera: Tricogrammatidae) to low concentrations of nicosulfuron atrazine increased adult emergence [39]. Additionally, feeding *Spodoptera eridania* (Lepidoptera: Noctuidae) hormetic doses of Cry1Ac soyabean leaves resulted in shorter development time from neonate to pupae [40]. Low-dose irradiation during the early life stages of *S. litura* in the present study increased longevity in both virgin and mated moths (0.75 Gy and 1.0 Gy administered to eggs, 1.0 Gy to larva and to pupae), but the increase in life span was more pronounced in virgin moths than in mated moths. This might be attributed to the additional cost of reproduction for mated moths. Interestingly, the mating success of *S.litura* was better in male moths derived from the hormetic low-dose irradiation treatments during the early ontogenetic stages [41].

The above indicates that low doses of gamma radiation administered to different ontogenetic stages could benefit both male and female moths, providing a significant increase in longevity and better survival rates. Similar results have been reported in the common fruit fly *Drosophila melanogaster* (Diptera: Drosophilidae), the codling moth *Cydia pomonella* (Lepidoptera: Tortricidae), and *Drosophila subobscura* (Diptera: Drosophilidae) when exposed to low doses of gamma radiation [42,43,44,45,46].

The mechanisms underlying stress factor-induced hormesis in arthropods remain largely unknown, despite a few recent advances [10,47,48]. Many researchers have explained the genetic regulation behind an insect’s lifespan. In our study, brain and fat tissues were used as focal tissues because the brain produces several important hormones and is also prone to many age-related degenerative changes [49]. Fat bodies in insects are analogous to vertebrate liver and adipose tissue and are also a production and processing site for many haemolymph proteins in adult moths [50]. Saunders and Verdin have demonstrated that exposing *C. elegans* to a variety of mild stress factors induced an adaptive biological response that increased lifespan. These were mediated by transcription factors that regulate both cellular stress response and lifespan [15]. The *Sir2* family has seven members which are termed as sirtuins (SIRTs), and *sirt1* has been reported to be activated by various stressors, including heat shock, hypoxia, misfolded proteins, free radicals, and adenosine triphosphate depletion, and it modulates lifespan by targeting the transcriptional regulators like *p53, NF-kB, HSF1, FOXO1, 3* and *4* [15]. A study on *D. melanogaster* showed that if the *Sir2* gene (orthologue of mammalian *sirt1*) was overexpressed in adult fat bodies, it could alone extend the longevity of both male and female flies [51]. Some studies also suggest that the *FOXO* protein family of *FOX* (Forkhead box) transcription factors might play an important role in protecting cells from stress, cell cycle arrest, metabolism, and tumor suppression [52]. In *C. elegans*, loss-of-function mutation in *daf-16/foxo* completely suppressed their longevity [53], and in *Drosophila*, overexpression of the *dFOXO* protein showed increase in lifespan via the insulin/insulin-like growth factor (IGF)-like signaling (IIS) pathway [54,55]. In our study, both *sirtuin 2 like* (encodes *sirt1*) and *foxo* genes showed a significant increase in the expression of mRNA levels after low-dose irradiation treatment. Such changes in gene expressions suggest that they might contribute to extend the lifespan of *S. litura*.

The *FOXO* protein targets antioxidant genes under oxidative stress, such as catalase (*cat*), superoxide dismutase (*sod*), and glutathione peroxidase (*gpx*) genes, which contribute to the detoxification of reactive oxygen species (ROS), repair of damaged DNA, and cell survival. Hence, it has been correlated with increased lifespan in many species [28]. Accumulating evidence from animal models also indicates that exposure to low doses of ionizing radiation may lead to the activation of pathways involved in endogenous antioxidant defense, such as superoxide dismutase, catalase, glutathione, glutathione reductase, and glutathione peroxidase, in different tissues including the brain, the liver, the spleen, and the pancreas, and it may cause stabilization of the ROS level [56,57]. Similar results were also obtained in the present study where *sod* and *cat* antioxidant genes showed an increased mRNA level in moths derived from low-dose irradiation during early ontogenetic stages, therefore potentially leading to the stabilization of ROS.

Additionally, ataxia telangiectasia mutated (*atm)* gene, which is a cellular DNA damage sensor, and *p53*, which plays a key role in DNA damage response caused by different stress agents such as heat, ionizing radiation, oxidation, and UV irradiation, might also contribute to regulating the longevity of an organism. These genes are important for initiating cellular protective processes such as double- and single-strand DNA breaks, cell cycle control, and apoptosis [58,59,60]. Autophagy occurs in response to heat stress, starvation, and oxidative stress [21,22,23,24,61,62], but it also participates in the removal of damaged and/or incorrectly functioning organelles and macromolecules [63,64]. It may be possible that stress resistance and longevity factors such as *Sirt1, mTOR, Foxo3, NF-jB*, and *p53* can regulate the ageing process via autophagy as well [65]. Some reports also have showed increased *atg8* gene levels under stress conditions, and *Foxo1, Foxo3*, and *E2F1* may help in the transcriptional regulation of *atg8* [66]. In our study, no significant difference was observed in the gene expression of *atm* and *p53* in response to hormetic gamma doses, which presumably could not express DNA damage sensor and apoptosis genes. The autophagy gene *atg8* showed upregulation in this study, as the autophagy plays a role in promoting programmed cell survival and might help in cell survival after exposing the insects to the hormetic gamma dose, which might induce radioprotective effects [67,68]. Some studies have also demonstrated similar results where low-dose (<10 μM) exposure to heavy metals like cadmium upregulated autophagy without the induction of apoptosis in proximal convoluted tubules (PCT) in vivo in rat kidneys [69].

The quality of the mass-reared insects to be released in sterile insect release programs is pivotal for the efficiency of these pest control programs. There is no literature on radiation hormesis of lepidopteran insects, although there are a few reports on hormesis with Lepidoptera using low doses of different stress agents. For instance, the preconditioning of adults of the cactus moths *Cactoblastis cactorum* (Berg) (Lepidoptera: Pyralidae), with a low dose of hormetins minimized the negative effect of high doses of radiation. One hour of exposure to anoxia (<0.1% oxygen) followed immediately by irradiation with 200 Gy enhanced the performance of the sterile male moths [70]. The anoxia conditioning could also alter the relationship between the radiation dose and the survival and sterility of *C. cactorum* [5]. A similar study was carried out on *Trichoplusia ni* (Lepidoptera: Noctuidae), where a low oxygen atmosphere enhanced the survival of moths post irradiation treatment [71]. Similarly, it increased radio tolerance when exposure to irradiation treatment was preceded by low oxygen treatment [6]. Hence, preconditioning with low doses of oxygen might be applied in insect management to improve the quality of the insects [72].

It is therefore concluded that the quality of irradiated insects can be improved by minimizing the somatic damage of high doses of radiation through preconditioning with low hormetic doses of ionizing radiation. This will be advantageous in management programs of lepidopteran pests that are using SIT or IS.

## 5. Conclusions

The findings presented here indicate that low doses of ionizing radiation administered to early ontogenetic stages could increase the life expectancy of laboratory reared *S. litura*, possibly due to the influence on longevity-associated gene factors such as *foxo, sirtuin 2 like/sirt1**, atg8, sod*, and *cat*. Several growth and development parameters of the preimaginal moth stages were positively influenced by low doses of radiation, indicating a low irradiation dose-induced hormetic effect. In addition to finding the appropriate dose, determining specific windows in the developmental period of the laboratory reared *S.litura* for low-dose irradiation treatment is equally important. As evidenced by our study, the different ontogenetic stages that received low doses of irradiation were suitable for exhibiting a hormetic effect in adult behavior. As per the findings of this study, radiation-induced hormesis could improve the quality of mass-reared *S. litura* moths, and it might, therefore, increase the efficiency of the SIT and F1 sterility programs. Furthermore, studies on ascertaining the effect of radiation hormesis on the mating competitiveness of the substerilized male moths (at 130 Gy) primed with low hormetic gamma radiation are in progress.

## Figures and Tables

**Figure 1 insects-13-00933-f001:**
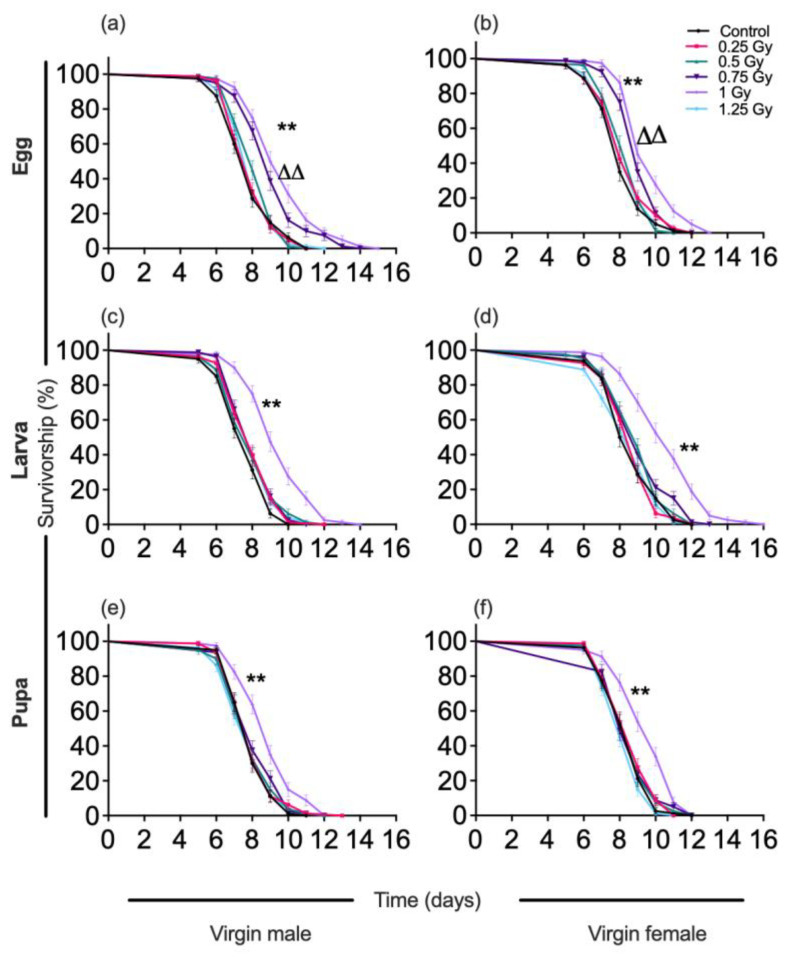
Survivorship curves of virgin adult male and female *Spodoptera litura* exposed to the low-dose ionizing radiation (LDIR) in different ontogenetic (Egg, Larva, Pupa) stages. (**a**) Male, (**b**) Female moths, irradiated in egg stage; (**c**) Male, (**d**) Female moths, irradiated in larval stage; (**e**) Male, (**f**) Female moths, irradiated in pupal stage. Significant difference between survival curves was conducted using a log-rank (Mantel-Cox) test for each condition, and significance is denoted by asterisks in triangles in 0.75 Gy and 1.0 Gy (ΔΔ *p* < 0.001 level, ** *p* < 0.001 level).

**Figure 2 insects-13-00933-f002:**
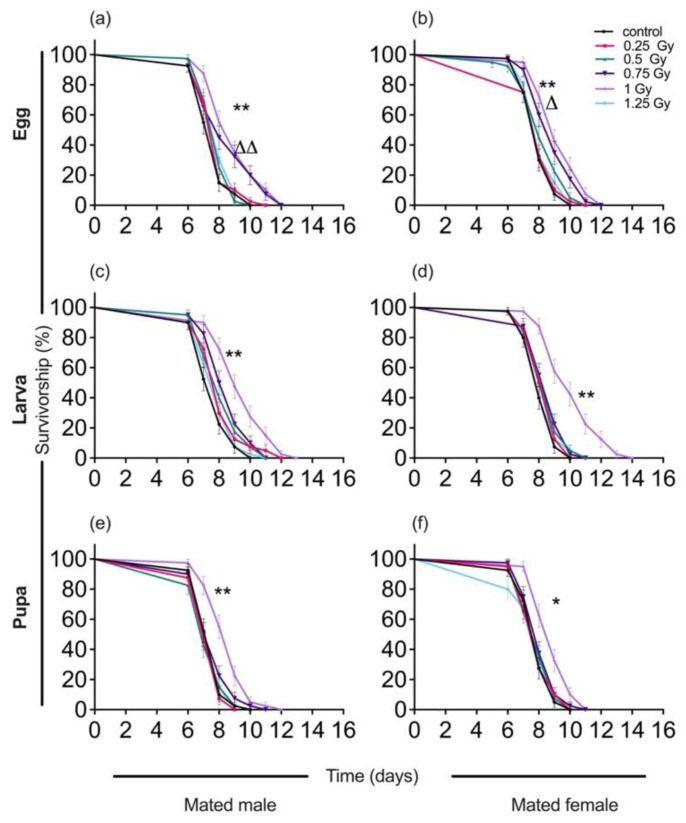
Survivorship curves of mated adult male and female *Spodoptera litura* exposed to low-dose ionizing radiation (LDIR) in different ontogenetic (Egg, Larva, Pupa) stages. (**a**) Male, (**b**) Female moths, irradiated in egg stage; (**c**) Male, (**d**) Female moths, irradiated in larval stage; (**e**) Male, (**f**) Female moths, irradiated in pupal stage. Significant difference between survival curves was conducted using a log-rank (Mantel-Cox) test for each condition, and significance is denoted by triangles in 0.75 Gy and asterisks in 1.0 Gy. (Δ *p* < 0.05, ΔΔ *p* < 0.001 level; * *p* < 0.05 level, ** *p* < 0.001 level).

**Figure 3 insects-13-00933-f003:**
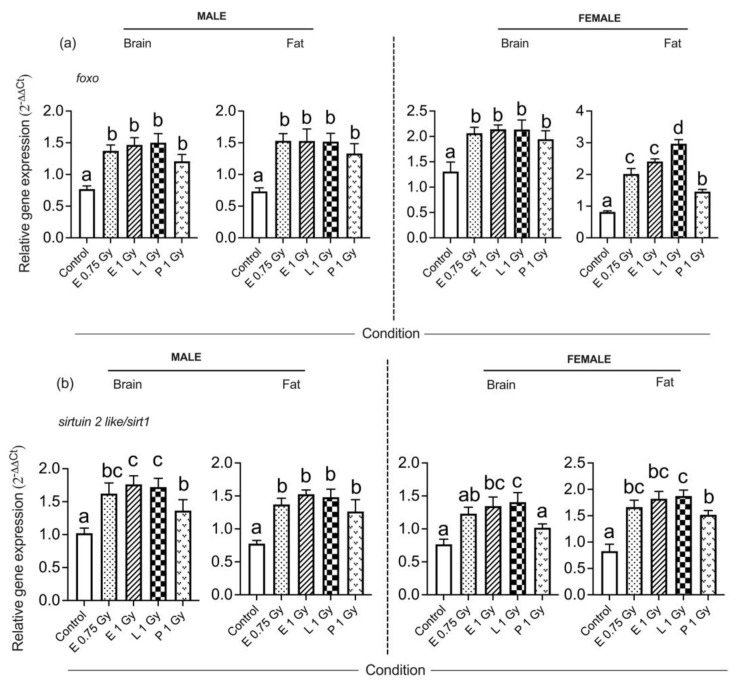
Profile of differentially expressed genes in adult male and female moths, *Spodoptera litura*, due to potential hormetic doses given in different ontogenetic stages. Genes include: (**a**) *foxo*, (**b**) *sirtuin 2 like/sirt1*, (**c**) *atm*, (**d**) *atg8*, (**e**) *sod*, (**f**) *cat* and (**g**) *p53*. Mean bars represented by the same letters for a specific gene expression in a particular tissue (brain or fat) of a specific sex (male or female) are not significantly different at *p* < 0.05. One-way ANOVA followed by Tukey’s multiple comparisons (*p* < 0.05) was considered significant. E-egg; L-larva; P-pupa.

**Table 1 insects-13-00933-t001:** Details of primer sequences used for the quantitative real-time PCR (qPCR).

S. No.	Gene	Primer Sequences	Accession No.
1	*EF1* α	F 5′-GACAAACGTACCCATCGAGAAG-3′ R 5′-GATACCAGCCTCGAACTCAC-3′	XM_022965580.1
2	*foxo*	F 5′-TGGTGGATGATCAATCCTGATG-3′ R 5′-CCTTCTTCTTTACTCTGCCTCTT-3′	XM_022969964.1
3	*sirtuin 2 like/sirt1*	F 5′-GCTCATCATCATGGGTTCCT-3′ R 5′-CCTGCCTTCTCACGGTTTAT-3′	XM_022962252.1
4	*atm*	F 5′-CCCTCACCACACATTACCTATC-3′ R 5′-CCCGCTCGCGTTCATTATTA-3′	XM_022978157.1
5	*atg8*	F 5′-GGAGTGCGCAATACCTCAA-3′ R 5′-CGGAACACGATCAGGATACTTC-3′	JX183217.1
6	*sod*	F 5′-GCCAACCTAGTTAGCATGTAAGA-3′ R 5′-ATGTGCCCTGCACCAATAA-3′	XM_022972232.1
7	*cat*	F 5′-TGAGACCAGAAACCACACATC-3′ R 5′-GTGCCACTCCTTGAGCATTA-3′	JQ663444.1
8	*p53*	F 5′-AGTCTAGCGACACCTCTAAGT-3′ R 5′-GTTCTCCTTCCCTTCCTGATTAC-3′	XM_022971404.1

**Table 2 insects-13-00933-t002:** (**a**): Effect of low-dose ionizing radiation (0.25–1.25 Gy) on growth features of *Spodoptera litura* (Fabr.) treated in egg stage. (**b**): Effect of low-dose ionizing radiation (0.25–1.25 Gy) on growth features of *Spodoptera litura* (Fabr.) treated as third instar larvae (L3). (**c**): Effect of low-dose ionizing radiation (0.25–1.25 Gy) on growth features of *Spodoptera litura* (Fabr.) treated in pupal stage.

**(a)**							
	**Control**	**0.25 Gy**	**0.5 Gy**	**0.75 Gy**	**1 Gy**	**1.25 Gy**	***F* Value**
Pupal Weight (g)	0.273 ± 0.004 a	0.267 ± 0.003 a	0.265 ± 0.002 a	0.287 ± 0.003 b	0.298 ± 0.003 c	0.262 ± 0.003 a	16.99 (5, 54)
Pupation (%)	88.6 ± 2.24 ab	83.2 ± 3.46 a	85.2 ± 3.85 ab	88.4 ± 2.71 ab	92.5 ± 1.91 b	83.5 ± 4.98 a	10.08 (5, 54)
Adult Emergence (%)	81.9 ±2.90 ab	75.3 ± 2.11 a	77.5 ± 4.75 a	88.5 ± 3.08 bc	91.2 ± 1.90 c	79.4 ± 2.70 a	9.17 (5, 54)
Development Period(Egg-Adult) (days)	27.3 ± 0.07 a	27.3 ± 0.06 a	27.5 ± 0.08 a	27.4 ± 0.07 a	26.8 ± 0.06 b	26.7 ± 0.09 b	12.03 (5, 54)
Growth Index	2.96 ± 0.11 ab	2.74 ± 0.13 a	2.81 ± 0.11 a	3.21 ± 0.14 bc	3.38 ± 0.16 c	2.87 ± 0.12 a	7.76 (5, 54)
**(b)**							
	**Control**	**0.25 Gy**	**0.5 Gy**	**0.75 Gy**	**1 Gy**	**1.25 Gy**	***F* Value**
Pupal Weight (g)	0.275 ± 0.003 a	0.27 ± 0.003 a	0.265 ± 0.003 a	0.26 ± 0.003 a	0.272 ± 0.003 a	0.275 ± 0.003 a	7.42 (5,54)
Pupation (%)	88.2 ± 2.95 ab	85.7 ± 2.35 a	87.5 ± 1.75 a	88.51 ± 1.95 ab	92.8 ± 1.91 b	84.50 ± 2.34 a	6.33 (5,54)
Adult Emergence (%)	82.7 ±2.8 ab	79.5 ± 2.2 a	82.3 ± 1.7 ab	84.9 ± 2.1 ab	88.9 ± 2.7 b	78 ± 2.6 a	3.82 (5,54)
Development Period(L3-Adult) (days)	12.9 ± 0.07 a	13.0 ± 0.06 ab	13.1 ± 0.07 b	12.4 ± 0.06 a	12.8 ± 0.06 a	13.0 ± 0.07 ab	2.29 (5,54)
Growth Index	6.33 ± 0.24 ab	6.11 ± 0.25 a	6.28 ± 0.27 ab	6.5 ± 0.26 ab	7.01 ± 0.22 b	6.01 ± 0.23 a	4.29 (5,54)
**(c)**							
	**Control**	**0.25 Gy**	**0.5 Gy**	**0.75 Gy**	**1 Gy**	**1.25 Gy**	***F* Value**
Adult Emergence (%)	83.5 ±1.9 a	81.7 ± 2.2 a	82.9 ± 1.8 a	79.5 ± 1.5 a	84.5 ± 1.9 a	79.7 ± 2.2 a	0.76 (5, 54)
Development Period(Pupa-Adult) (days)	7.4 ± 0.05 a	7.5 ± 0.07 a	7.4 ± 0.07 a	7.4 ± 0.06 a	7.4 ± 0.06 a	7.4 ± 0.06 a	0.62 (5,54)
Growth Index	11.24 ± 0.33 a	10.8 ± 0.42 a	11.10 ± 0.40 a	10.71 ± 0.32 a	11.39 ± 0.34 a	11.05 ± 0.49 a	0.90 (5,54)

Means followed by the same letters within a row are not significantly different at *p* < 0.05. One-way ANOVA followed by Tukey’s multiple comparisons (*p* < 0.05) was considered significant. The percentage data was arcsine transformed before ANOVA analysis. (**a**) In each replicate, a cohort of 60–70 eggs was studied for % pupation and % adult emergence. The average of the pupal weight and developmental period of ten insects from each cohort was computed as one replicate. Growth Index = % Adult emergence/Developmental period (days). (**b**) In each replicate, a cohort of 50–60 L3 was studied for % pupation and % adult emergence. The average of the pupal weight and developmental period of ten insects from each cohort was computed as one replicate. Growth Index = % Adult emergence/Developmental period (days). (**c**) In each replicate, a cohort of 25–30 pupae (2–3 days old) was studied for % pupation and % adult emergence. The average of the pupal weight and developmental period of ten insects from each cohort was computed as one replicate. Growth Index = % Adult emergence/Developmental period (days).

**Table 3 insects-13-00933-t003:** Effect of low-dose ionizing radiation (LDIR) on the longevity of *Spodoptera litura* treated in various ontogenetic stages.

	Control	0.25 Gy	0.5 Gy	0.75 Gy	1 Gy	1.25 Gy	*F* Value
**LDIR administered to Egg**							
Virgin Male	7.9 ± 0.16 a	8.1 ± 0.14 a	8.3 ± 0.13 a	9.2 ± 0.19 b	9.7 ± 0.21 b	8.1 ± 0.13 a	20.45 (5, 54)
Virgin Female	8.1 ± 0.15 a	8.3 ± 0.17 a	8.4 ± 0.12 a	9.1 ± 0.13 b	9.7 ± 0.16 c	8.3 ± 0.16 a	16.89 (5,54)
Mated Male	7.7 ± 0.16 a	7.9 ± 0.17 a	7.9 ± 0.13 a	8.6 ± 0.28 b	9.1 ± 0.25 b	7.9 ± 0.17 a	6.84 (5, 54)
Mated Female	8.1 ± 0.15 a	8.2 ± 0.16 a	8.4 ± 0.22 a	9.0 ± 0.21 b	9.4 ± 0.21 b	8.3 ± 0.16 a	8.07 (5, 54)
**LDIR administered to Larva**							
Virgin Male	7.7 ± 0.14 a	8.1 ± 0.15 a	8.0 ± 0.17 a	8.2 ± 0.14 a	9.6 ± 0.18 b	8.1 ± 0.15 a	17.24 (5, 54)
Virgin Female	8.7 ± 0.16 ab	8.7 ± 0.15 ab	9.1 ± 0.17 ab	9.2 ± 0.19 b	10.7 ± 0.22 c	8.6 ± 0.18 a	18.31 (5, 54)
Mated Male	7.7 ± 0.17 a	8.1 ± 0.22 a	8.2 ± 0.21 a	8.6 ± 0.21 b	9.6 ± 0.25 c	7.9 ± 0.18 a	9.62 (5, 54)
Mated Female	8.2 ± 0.14 a	8.4 ± 0.15 a	8.5 ± 0.17 a	8.7 ± 0.16 a	10.2 ± 0.27 b	8.4 ± 0.17 a	16.03 (5, 54)
**LDIR administered to Pupa**							
Virgin Male	8.0 ± 0.12 a	8.1 ± 0.15 a	7.9 ± 0.15 a	8.2 ± 0.15 a	9.0 ± 0.17 b	7.8 ± 0.16 a	7.32 (5, 54)
Virgin Female	8.5 ± 0.14 a	8.6 ± 0.14 a	8.6 ± 0.15 a	8.6 ± 0.15 a	9.5 ± 0.172 b	8.3 ± 0.12 a	8.87 (5, 54)
Mated Male	7.5 ± 0.13 a	7.4 ± 0.13 a	7.4 ± 0.16 a	7.7 ± 0.18 a	8.6 ± 0.19 b	7.5 ± 0.14 a	8.78 (5, 54)
Mated Female	7.9 ± 0.15 a	8.0 ± 0.16 a	8.0 ± 0.17 a	8.2 ± 0.17 a	8.9 ± 0.17 b	7.8 ± 0.19 a	6.02 (5, 54)

Means followed by the same letters within a row are not significantly different at *p* < 0.05 level. One-way ANOVA followed by Tukey’s multiple comparisons (*p* < 0.05) was considered significant. *n* = 10; The average longevity of ten insects (virgin and mated) from each cohort was computed as one replicate.

## Data Availability

All data are contained within the article.

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
