# Peer review of "Radiation Hormesis to Improve the Quality of Adult Spodoptera litura (Fabr.)"

_insects, 2022, doi:10.3390/insects13100933_

Round 1

Reviewer 1 Report

In this manuscript, Vimal and colleagues investigated whether the application of low doses of ionizing radiation on various ontogenetic stages of Spodoptera litura could have a stimulating (hormetic) effect on growth and viability. The results are clear indicating that gamma irradiation doses as low as 0.75-1.0 Gy to eggs, larvae and pupae could have a positive effect on pupal weight, adult emergence, and growth index. In addition, they showed that these low hermetic doses upregulate the expression of genes playing a role in longevity and viability / anti-oxidative pathways such as foxo, sirtuin 2 like/sirt1, atg8, cat and sod. This study is an important contribution to the area of insect pest control because it shows that the application of hormetic doses of ionizing radiation at early developmental stages may positively affect the productivity and biological quality of sterile insects thus enhancing the efficacy and cost-effectiveness of sterile insect technique (SIT) applications. I have the following comments which the authors may consider to further improve their manuscript:

1) My major comment has to do with the Introduction and the Discussion where the authors have ignored similar studies in other insect species targeted by SIT (see publications by Prof. Hahn’s lab) or key studies / recent reviews about hormesis in insects (see recent reviews by Prof. Cutler’s lab or by Dr. Coval and colleagues – two of them in 2022). The authors may consider updating these sections to include these studies.

2) Lines 31-32: as “radiation hormesis” and “Spodoptera litura” are in the title, they are not needed in the Keywords.

3) Line 43: delete 100%.

4) Line 74: Sirt1 and sir2 should be in italics. As all gene names should be in italics, please check this throughout the manuscript.

5) Line 136: space between “with” and “a cotton”. There are many such cases of extra spacing or lack of spacing throughout the manuscript, please check thoroughly.

6) Line 141: replace “paired” with “put or placed”.

7) Line 158: is “0.1% agarose gel” correct?

8) Line 451: incomplete reference.

Author Response

Response to Reviewer 1 : Comments and Suggestions for Authors

Remarks: In this manuscript, Vimal and colleagues investigated whether the application of low doses of ionizing radiation on various ontogenetic stages of Spodoptera litura could have a stimulating (hormetic) effect on growth and viability. The results are clear indicating that gamma irradiation doses as low as 0.75-1.0 Gy to eggs, larvae and pupae could have a positive effect on pupal weight, adult emergence, and growth index. In addition, they showed that these low hermetic doses upregulate the expression of genes playing a role in longevity and viability /anti-oxidative pathways such as foxosirtuin 2 like/sirt1atg8cat and sod. This study is an important contribution to the area of insect pest control because it shows that the application of hormetic doses of ionizing radiation at early developmental stages may positively affect the productivity and biological quality of sterile insects thus enhancing the efficacy and cost-effectiveness of sterile insect technique (SIT) applications. I have the following comments which the authors may consider to further improve their manuscript:

Response: The authors express sincere thanks for the appreciation of the Reviewer for this manuscript. The responses to the various queries made by the Reviewer are as follows. 

Query: 1) My major comment has to do with the Introduction and the Discussion where the authors have ignored similar studies in other insect species targeted by SIT (see publications by Prof. Hahn’s lab) or key studies / recent reviews about hormesis in insects (see recent reviews by Prof. Cutler’s lab or by Dr. Coval and colleagues – two of them in 2022). The authors may consider updating these sections to include these studies.

Response:  The necessary changes are made.  As sterilizing or sub-sterilizing doses of gamma radiation have been the subject of extensive research, little attention has been paid to the effects of extremely low radiation doses inducing hormesis in the context of Lepidopteran insects that would be exposed to higher gamma doses later for the inducing reproductive sterility. Although , the concept of mild stress-induced physiological hormesis has been explored towards management of some Lepidopteran insects using low dose of oxygen atmosphere (Martinez 2016, Condon 2017).

Query: 2) Lines 31-32: as “radiation hormesis” and “Spodoptera litura” are in the title, they are not needed in the Keywords.

Response:  The necessary changes are made.

Query: 3) Line 43: delete 100%.

Response:  The necessary changes are made.

Query: 4) Line 74: Sirt1 and sir2 should be in italics. As all gene names should be in italics, please check this throughout the manuscript.

Response:  The necessary changes are made, and the whole manuscript is checked accordingly.

Query: 5) Line 136: space between “with” and “a cotton”. There are many such cases of extra spacing or lack of spacing throughout the manuscript, please check thoroughly.

Response:  The necessary changes are made.

Query: 6) Line 141: replace “paired” with “put or placed”.

Response:  The necessary changes are made.

Query: 7) Line 158: is “0.1% agarose gel” correct?

Response:  Yes, it was written by mistake, and it is rectified  now as 1% agarose gel.

Query: 8) Line 451: incomplete reference.

Response:  The necessary changes are made.

____________

Reviewer 2 Report

The manuscript “Radiation hormesis to improve the quality of adult Spodoptera litura (Fabr.)” assesses the extent to which radiation hormesis improves the growth and viability of egg, larval, and pupal moths. I consider this a relevant study with the potential to be used in operational SIT programs after large-scale validation of the authors’ results. However, I have concerns regarding the lack of information about radiation treatments. The authors do not mention anything about the dosimetry system used, the uncertainty of the dosimetry system, absorbed dose, and DUR in the study. I found the results and discussion well written. However, I cannot provide a full review of these sections without evaluating the quality of the radiation treatments conducted by the authors. Without an accurate description of the radiation treatments, the readers cannot assess the validity of the findings reported in the paper. Only minor corrections should be made in the introduction. Methods should be revised to address the questions raised in my comments. Below are specific comments that may be helpful to the authors.

Lines 55 and 56: the authors should review this sentence because the statement about the complete lack of attention to hormesis in Lepidoptera is not accurate. Several types of research evaluate the extent to which low oxygen hormetic conditioning affects the radiotolerance of lepidopteran species. Below is a list of studies on the topic:

  • https://doi.org/10.1111/eva.13141https://journals.flvc.org/flaent/article/view/88670
  • https://doi.org/10.1653/024.099.sp113
  • https://doi.org/10.1603/EC13370
  • https://doi.org/10.1002/ps.5768
  • https://doi.org/10.1093/jee/tow273

Line 102: consider adding a space between letters and symbols. I would suggest using the multiplication sign “ × ” instead of the letter “ x ” for measurements. Note that other manuscript sections should be reviewed but were not specified here (e.g., line 115).

Lines 108 to 115: the authors should provide information about the dosimetry system used to measure the absorbed dose of the radiation treatments and the uncertainty of their dosimetry system. Reporting only the nominal doses of the treatments may not be sufficient for evaluating their validity by the readers. I suggest the authors review the IAEA guidelines for dosimetry: https://www.iaea.org/resources/manual/dosimetry-for-sit-standard-operating-procedures-for-gafchromictm-film-dosimetry-system-for-gamma-radiation-v10 

Line 130: the authors should add a space between “3” and “day”. The same comment applies to other parts of the manuscript. Please check for the typo across the manuscript.

Line 135: please provide the cage measurements used in the longevity tests with virgin insects.

Author Response

Response to Reviewer 2:  Comments and Suggestions for Authors

Remarks: The manuscript “Radiation hormesis to improve the quality of adult Spodoptera litura (Fabr.)” assesses the extent to which radiation hormesis improves the growth and viability of egg, larval, and pupal moths. I consider this a relevant study with the potential to be used in operational SIT programs after large-scale validation of the authors’ results. However, I have concerns regarding the lack of information about radiation treatments. The authors do not mention anything about the dosimetry system used, the uncertainty of the dosimetry system, absorbed dose, and DUR in the study. I found the results and discussion well written. However, I cannot provide a full review of these sections without evaluating the quality of the radiation treatments conducted by the authors. Without an accurate description of the radiation treatments, the readers cannot assess the validity of the findings reported in the paper. Only minor corrections should be made in the introduction. Methods should be revised to address the questions raised in my comments. Below are specific comments that may be helpful to the authors.

Response: The authors express sincere thanks for the appreciation and the relevant remarks of the Reviewer for this manuscript. The responses to the various queries made by the Reviewer are as follows. 

Query: Lines 55 and 56: the authors should review this sentence because the statement about the complete lack of attention to hormesis in Lepidoptera is not accurate. Several types of research evaluate the extent to which low oxygen hormetic conditioning affects the radiotolerance of lepidopteran species. Below is a list of studies on the topic:

  • https://doi.org/10.1111/eva.13141
  • https://journals.flvc.org/flaent/article/view/88670
  • https://doi.org/10.1653/024.099.sp113
  • https://doi.org/10.1603/EC13370
  • https://doi.org/10.1002/ps.5768
  • https://doi.org/10.1093/jee/tow273

Response:  The necessary information related to  most of these papers has been added in the text, which has highlighted the role of hormesis, although there is a scanty literature available on the radiation hormesis in case of Lepidopteran insects,  

Query: Line 102: consider adding a space between letters and symbols. I would suggest using the multiplication sign “ × ” instead of the letter “ x ” for measurements. Note that other manuscript sections should be reviewed but were not specified here (e.g., line 115)

Response:  The appropriate  space and  symbol “ × ”  are introduced.

Query: Lines 108 to 115: the authors should provide information about the dosimetry system used to measure the absorbed dose of the radiation treatments and the uncertainty of their dosimetry system. Reporting only the nominal doses of the treatments may not be sufficient for evaluating their validity by the readers. I suggest the authors review the IAEA guidelines for dosimetry: https://www.iaea.org/resources/manual/dosimetry-for-sit-standard-operating-procedures-for-gafchromictm-film-dosimetry-system-for-gamma-radiation-v10 

Response:  We agree with the comments of the referee. The necessary changes made as follows: The irradiator beam was calibrated following IAEA’s TRS-398 protocol, using calibrated Farmer type, 0.6 cc volume, ionization chamber in water phantom. During experimental phase, the measured machine output (dose rate) ranged from 1.09 Gy/min to 0.74 Gy/min. The dose uniformity was better than ±3% within the central 80% of the field and the overall output calibration uncertainty was within ±2%.

Query: Line 130: the authors should add a space between “3” and “day”. The same comment applies to other parts of the manuscript. Please check for the typo across the manuscript.

Response:  The necessary changes are made.

Query: Line 135: please provide the cage measurements used in the longevity tests with virgin insects.

Response:  The necessary changes are made.

____________
